# Local Tranexamic Acid for Preventing Hemorrhage in Anticoagulated Patients Undergoing Dental and Minor Oral Procedures: A Systematic Review and Meta-Analysis

**DOI:** 10.3390/healthcare10122523

**Published:** 2022-12-13

**Authors:** Asma Zaib, Muhammad Shaheryar, Muhammad Shakil, Azza Sarfraz, Zouina Sarfraz, Ivan Cherrez-Ojeda

**Affiliations:** 1Department of Research, University Medical & Dental College Faisalabad, Faisalabad 38800, Pakistan; 2Department of Research, Rawal Institute of Health Sciences, Islamabad 45550, Pakistan; 3Department of Research, Frontier Medical & Dental College, Abbottabad 22030, Pakistan; 4Department of Pediatrics and Child Health, The Aga Khan University, Karachi 74800, Pakistan; 5Department of Research and Publications, Fatima Jinnah Medical University, Lahore 54000, Pakistan; 6Department of Allergy and Pulmonology, Universidad Espíritu Santo, Samborondón 092301, Ecuador

**Keywords:** dental care, tranexamic acid, vitamin K antagonists, direct oral anticoagulants, mouthwash, postoperative bleeding

## Abstract

Dental procedures have posed challenges in managing anticoagulated patients due to early reports of oral hemorrhage. This study aims to evaluate the risks of postoperative bleeding with the local application of tranexamic acid. A systematic search was conducted until 31 March 2022, with keywords including tranexamic acid, oral hemorrhage, dental, and/or coagulation. The following databases were searched: PubMed, Scopus, Web of Science, CINAHL Plus, and Cochrane Library. Statistical analysis was conducted using Review Manager 5.4. In total, 430 patients were pooled in with the local application of tranexamic acid using mouthwash, irrigation, and compression with a gauze/gauze pad. The mean age was 61.8 years in the intervention group and 58.7 in the control group. Only 4 patients in the intervened group out of the 210 discontinued the trial due to non-drug-related adverse events. The risk difference was computed as −0.07 (*p* = 0.05), meaning that patients administered with local antifibrinolytic therapy for postoperative bleeding reduction for dental procedures were at a 7% less risk of oral bleeding. Current evidence on managing anticoagulated patients undergoing dental or oral procedures remains unclear. The present study presents favorable outcomes of postoperative bleeding with local tranexamic acid used in the postoperative period.

## 1. Introduction

Anticoagulation regimens are administered to patients at risk of thromboembolism, such as acute coronary syndrome, atrial fibrillation (AF), or prosthetic cardiac valves (CV) [1,2,3,4]. Dental or minor oral procedures have posed challenges in managing anticoagulated patients due to early reports of major oral bleeding in these patients [2,3,4,5,6]. A standardized approach maintains a therapeutic range of international normalized ratio (INR) for controlling hemorrhage risk [7]. In routine practice, patients on anticoagulation undergoing dental or oral procedures continue with their anticoagulation regimen, and the INR is maintained at the desired range, albeit with unclear scientific evidence in favor of this practice [8]. The number and severity of bleeding events also depend on surgery-related characteristics such as the size of the wound and the extent of invasive procedures [9]. Further, the use of a sutureless technique after tooth extraction influences postoperative bleeding, as well as possible infection of the wound [10,11,12]. In dental or minor oral surgeries, the risk of bleeding is anatomically high, and the use of anticoagulants disturbs the salivary enzymes and tissue plasminogen activator (t-PA) balance that maintains the hemostatic effect [8]. Individuals on antithrombotic medications form a priority group due to their high risk of bleeding after dental or oral procedures [5]. While the evidence is limited, it is unnecessary to alter anticoagulant therapy in most patients as the risk of stopping or reducing these medications is higher than using local measures to control bleeding after more invasive dental procedures [13]. In patients at increased risk of bleeding, such as anticoagulated patients with underlying chronic health conditions undergoing more extensive dental procedures, further research is warranted to establish management strategies [14,15,16,17,18,19]. 

This systematic review and meta-analysis aimed to evaluate high-grade evidence (i.e., placebo-controlled trials) that supports the local application of tranexamic acid, an antifibrinolytic, in reducing postoperative bleeding complications.

## 2. Materials and Methods

The protocol for this systematic review and meta-analysis was registered with Open Science Framework (OSF): osf.io/5rj49. 

### 2.1. Search Strategy and Study Selection

A systematic search was conducted until 31 March 2022, adhering to the Preferred Reporting Items for Systematic Reviews and Meta-Analyses (PRISMA) 2020 statement guidelines [20]. A combination of the following keywords was used: tranexamic acid, oral hemorrhage, dental, and/or coagulation. The Boolean logic was applied (and/or). PubMed, Scopus, Web of Science, CINAHL Plus, and Cochrane Library were searched. The inclusion criteria comprised randomized controlled trials that explored the effects of local tranexamic acid in preventing oral hemorrhage in anticoagulated patients of any gender undergoing dental procedures. These included randomized placebo-controlled trials employing an interventional and a control group (CG), with no time or language restrictions. Any non-English study was to be translated into the English language using Google Translate. Cohort studies, case series, case reports, meta-analyses, systematic reviews, brief reports, and letters to editors were excluded. The bibliographic entries were entered into EndNote X9 (Clarivate, Windows, Reference Management, USA) for managing and de-duplicating studies. The reference lists of all screened articles were reviewed to locate additional randomized controlled trials (umbrella methodology). 

The PRISMA flowchart showcasing the study selection is shown in Figure 1. In total, 2337 studies were identified from the databases, from which 352 duplicates were removed. All 1985 studies were screened, and of them, 1931 did not meet the inclusion criteria. Therefore, 54 studies were assessed for eligibility, and 49 of these were removed as they met the exclusion criteria. In the inclusion phase, 5 RCTs were added to the final analysis.

All investigators screened the titles and abstracts before a consensus was reached for inclusion in the systematic review and meta-analysis. Any disagreements were resolved with active discussion among all investigators. An a priori methodology was adopted using a Delphi process to ensure that the outcomes of interest and findings were enlisted.

### 2.2. Outcomes

The primary outcome was events of postoperative bleeds with the use of tranexamic acid. The dental procedures involved the teeth, palate, floor of the mouth, tongue, and other soft tissues in the oropharyngeal cavity. The secondary outcome was to provide an overview of adverse events due to stopping anticoagulation for dental extraction.

### 2.3. Data Analysis

The data was extracted onto a shared spreadsheet with the following information: (1) mean age (in years), (2) gender (male), (3) dosage and route of administration of local tranexamic acid, (4) indication for anticoagulation, (5) anticoagulation agent used and window before the last dose, (6) the number of patients with postoperative bleeds, and (7) adverse events due to stopping anticoagulation for dental extraction. The metric to assess the meta-analytical findings was the computation of risk difference (RD) among the two groups, for which a forest plot was generated. These data were thereby presented as RD with the I^2^ index to assess for the heterogeneity between the included studies. The *p*-value was considered to be statistically significant if it was less than or equal to 0.05. The Q test was also conducted to assess homogeneity among the pooled studies. The meta-analysis was conducted using Review Manager (RevMan 5.4; Cochrane). A funnel plot was also generated for all the RCTs that were pooled in this analysis to visually inspect for publication bias.

### 2.4. Risk of Bias Assessment

Version 2 of the Cochrane risk-of-bias tool for randomized trials (RoB 2) was used to assess the risk of bias in the included RCTs [21]. The RoB 2.0 assessment targets the following 5 domains. These include (1) biases present during the randomization process, (2) biases that arise due to deviations from intended interventions, (3) biases at any stage due to missing outcome data, (4) biases in the measurement of outcomes, and (5) biases in selecting reported results. The first component of this assessment was presented in the form of a traffic light plot of a study-by-study assessment. The second component was presented in an overall bias plot where domain-level judgments about the risk of bias were calibrated. They were classified as (1) low risk of bias, (2) some concerns, and (3) high risk of bias.

### 2.5. Role of Funding and Ethical Approval

No external funding was obtained for this study. Ethical approval was not required as only secondary clinical data were utilized for this meta-analysis. 

## 3. Results

The characteristics of anticoagulated patients included in this study are enlisted in Table 1. Ockerman et al., 2021, Queiroz et al., 2018, Soares et al., 2015, Borea et al., 1993, and Ramstrom et al., 1993, were all placebo-controlled, randomized trials. The total number of patients analyzed included 210 in the antifibrinolytic group and 220 in the control group (CG), pooling in 430 patients. The modality of drug administration included the local application of 10% tranexamic acid mouthwash once before dental extraction and 3 times a day for 3 days after that (Ockerman et al., 2021); irrigation and compression with gauze soaked in tranexamic acid (5%) and suture for a minimum of 5 min (Queiroz et al., 2018); gauze pad soaked in 4.8% tranexamic acid applied to the surgical alveolus for 8 min under biting pressure (Soares et al., 2015); 5% tranexamic acid as a mouthwash for 2 min 4 times a day for 7 days (Borea et al., 1993); surgical irrigation with 10 mL of 4.8%% tranexamic acid solution before suturing followed by mouthwash with 4.8% tranexamic acid solution for 2 min 4 times a day for 7 days, and application of a gauze pad soaked in the solution applied to the bleeding site for 20 min under biting pressure (Ramstrom et al., 1993). 

The overall average age of patients in the intervention group was 61.8 years, and in the placebo group, it was 58.7 years. The indications for Ockerman were AF (83% vs. 78.6%) and VTE (10.4% vs. 12.5%); Queiroz reported indications of CV (48.6%), DVT (40.5%), and CVA (10.8%). Soares had indications of mitral valve (MV) prolapse (47.4%), CV (23.7%), VTE (21.1%), and PE (5.2%). Borea enrolled patients who had CV (100%), and Ramstrom enrolled patients with CV (6.5% vs. 6.4%), VTE (34.8% vs. 27.7%), and CVD (13% vs. 19.6%). All the studies enrolled patients on warfarin and other coumarin drugs, except for Ockerman et al., whose patients took direct oral anticoagulants (DOACs). Overall, the postoperative bleeding outcomes were slightly better in the intervened group; the meta-analytical results are referred to in the subsequent section. In total, 4 patients in this cohort discontinued the trial post-enrollment, not due to the intervention drug side effects but overall health (Table 1).

All 5 of the 5 studies pooling in a total of 430 patients reported data on postoperative bleeding outcomes. The findings favored the local application of antifibrinolytics among the entire cohort of included patients. The results were as follows: RD = −0.07, 95% CI = −0.14, 0, *p* = 0.05. Patients being administered local antifibrinolytic therapy for postoperative bleeding reduction for dental procedures are at a 7% less risk of oral bleeding. 

Moderate heterogeneity was present in the included studies, with an I^2^ value of 58%. The Chi^2^ value was 9.43 (df = 4, *p* = 0.05), indicating that there was less variation across the studies as compared to within-subjects; the underlying null hypothesis is assumed where the antifibrinolytic treatment effect was the same across studies and variations were simply caused by chance (Figure 2).

A subgroup analysis was conducted among the cohort of patients. The following results were yielded on retesting the postoperative bleeding effects by removing the study with the highest weight (Ockerman et al., 2021): RD= −0.12, 95% CI = −0.2, −0.04 *p* = 0.002). The risk difference was 12%, meaning that the local antifibrinolytic group had a 12% lower risk of postoperative oral bleeding. Another sensitivity test was conducted by removing Soares et al., 2015 and Borea et al., 1993. The results were as follows: RD = −0.09, 95% CI = −0.17, 0, *p* = 0.04. The findings were comparable to the original findings, with the current test revealing a risk difference of 9%.

On visually inspecting the funnel plot, all five trials were well within the remit of an inverted funnel shape. Each dot represents a single study included in the meta-analysis. The x-axis shows the result of the study, whereas the y-axis represents the standard error of the effect estimate. Larger studies are placed toward the top, while the lower-powered studies are placed at the bottom. With an ideal shape of an inverted funnel or pyramid, the scatter is present due to sampling variation. The shape seen in this funnel plot is expected, given the wide range of standard errors. Publication bias may overall be minimal, although our findings must be interpreted with caution (Figure 3). 

On noting the bias arising from the randomization process in the five studies, three studies had low concerns, whereas two studies had some concerns. On assessing biases due to deviations from the intended interventions, four studies had a low risk of bias, whereas one study had some concerns. When assessing bias due to missing outcome data, three studies had low concerns, one had a high risk of bias, and one had some concerns. The biases in missing outcome data pertained to unobtainable information for outcomes of interest in this study. On noting bias in the measurement of the outcome, three studies had some concerns, whereas two studies had low concerns. For bias in the selection of the reported result, four studies had low concerns, whereas one had some concerns. Overall, two studies had some concerns, whereas three studies had low concerns (Figure 4).

## 4. Discussion

Our results indicate that, among anticoagulated patients, the use of local tranexamic acid reduces the risk of postoperative bleeding. This effect is significant for 4.8–10% tranexamic acid locally as a mouthwash for 1–2 min 3–4 times a day for 3–7 days, or irrigation and compression with soaked gauze at the surgical site for 8–20 min as needed. Of the indications for anticoagulation in our findings, the most common were atrial fibrillation (AF), venous thromboembolism (VTE), and other cardiovascular diseases (CVDs). Our results demonstrate a significant risk reduction of 7% using local tranexamic acid to control postoperative bleeding in anticoagulated patients undergoing dental procedures. While the overall risk of dental procedures identified in the included studies is low, this risk is still higher among anticoagulated patients, as shown in previous studies [27,28]. The most commonly used antifibrinolytic agent is tranexamic acid. It has been incorporated into the list of essential medications, albeit with unclear guidelines on the initiation and duration of local antifibrinolytics in the specific context of dental procedures among anticoagulated patients [29,30]. Tranexamic acid binds to plasminogen and subsequently inhibits the lysis of fibrin. When administered orally, tranexamic acid is not detected in saliva; however, local administration of tranexamic acid with gauze or as a mouthwash results in high salivary concentrations [31]. Therefore, the use of tranexamic acid, either as a mouthwash, or local application with gauze, may be considered a potent local hemostatic method in anticoagulated patients undergoing dental or oral procedures [32,33]. 

In the present study, direct oral anticoagulants (DOAC) and vitamin K antagonists (VKA) were used in patients undergoing dental procedures. In both types of anticoagulants, there is a risk of bleeding complications, and interruption of anticoagulant therapy before dental or oral surgery has been an issue of great controversy [34,35]. The decision to suspend these drugs may pose an increased risk of thromboembolism, whereas maintenance may result in postsurgical bleeding [36]. VKAs are currently the most commonly used anticoagulant drugs, e.g., warfarin, that act via inhibition of carboxylation of the vitamin K-dependent coagulation factors II, VII, IX, and X as well as protein C and S [8]. VKAs are difficult to control as this drug class has a long plasma half-life (e.g., warfarin, ~40 h) and depends on multiple hepatic enzyme systems for metabolism that interact with other medications and food [37]. Contrarily, DOACs are designed to target specific single enzymes across the coagulation cascade [38]. With comparable efficacy to VKA, DOACs have lower complications and shorter half-lives (9–13 h) [39]. However, for low-risk dental or oral procedures, both VKAs such as warfarin and DOACs may be safely continued unless the patients are on multiple medications or with renal insufficiency [40]. In the context of bleeding risk following dental procedures, there are mixed reports that DOACs are safer to use than VKAs [41,42], with data supporting better outcomes of post-procedural bleeding with DOACs and concomitant use of local hemostatic agents [43,44]. 

The difference between VKAs and DOACs is the facilitation of bridging or short-term interruption with the additional risk of relapsing thrombosis with DOACs [8]. However, as patients already on respective regimens may require oral procedures, it is of interest to understand the risk of interruption versus continuation of these medications in light of risks and benefits, specifically in DOACs, as there are only limited studies in these patients in the current literature [40,45]. Guidelines do not recommend the interruption of low-risk dental procedures, suggesting local hemostatic measures [5]. Regardless, optimizing local hemostatic methods is needed, as these are being explored. Specific measures include local surgical sutures, fibrin glue, and local antifibrinolytics (tranexamic acid or e-aminocaproic acid) in case of bleeding complications [46]. In patients on anticoagulation, these measures have also been conducted in the preoperative period, such as using tranexamic acid as a mouthwash prior to the oral procedure [47]. Considering the risk of bleeding complications with such dental or oral procedures, it is relevant to consider additional hemostasis intervention to determine the specific dosing, regimen, and selective efficacy of each method alone and in combination. 

### 4.1. Key Pharmacological Interactions of Tranexamic Acid

The current literature reports that tranexamic acid ought not to be used in case disseminated intravascular coagulation (DIC) or an active intravascular clotting process is present without the concurrent use of heparin due to the possibility of thromboembolism [48]. When tranexamic acid was used as a local treatment during dental procedures, patients with hemophilia reported a reduction of postoperative bleeding without increasing significant risks [49]. 

Tranexamic acid must be administered with caution given possible interaction with anticoagulants, including heparin and warfarin, and anti-bleeding medications, including anti-inhibitor coagulant concentrate and factor IX complex [50,51]. Other interactions may include hormonal birth control products, including patches and pills, estrogens, and tibolone [50]. While low-dose aspirin has not been associated with adverse events with tranexamic acid, non-steroidal anti-inflammatory drugs (NSAIDs), including ibuprofen, aspirin and naproxen, must be administered with caution [50,52].

### 4.2. Strengths and Limitations

The present study obtained data through rigorous literature searches that incorporate relevant randomized controlled trials. The results obtained pool together patients receiving treatment with VKAs and DOACs on therapeutic ranges ascertained by INR levels. There was mild publication bias with respect to the trials included in the study. The study with the largest effect size was removed to analyze the effect of local tranexamic acid on postoperative bleeding, which yielded a similarly positive outcome in favor of intervention. As DOACs are a newer class of drugs compared to VKAs, their beneficial effect with local tranexamic acid has been established. A limitation of this study was the moderate number of trials conducted, and the overall quality of evidence may be considered moderate. Additionally, while the efficacy of antifibrinolytic therapy has been established, there is a lack of robust certainty regarding its benefits over other measures of local hemostasis. Lastly, the lack of discontinuation of VKAs or DOACs was seen in four of the five trials, yet there is no clear conclusion, especially with DOACs, as these drugs have different half-lives and mechanisms of action compared to the well-known VKAs. 

## 5. Conclusions

This systematic review and meta-analysis present favorable local tranexamic acid outcomes in anticoagulated patients undergoing dental procedures when used post-operatively. However, given that current scientific evidence is unclear, with no consensus on managing anticoagulated patients undergoing dental procedures, our findings must be used with caution. Many individual-level and surgery-level characteristics influence the hemorrhage risk; it is relevant to consider add-on topical tranexamic acid to prevent postoperative complications. Further research is also required to confirm the benefit of local tranexamic acid compared to alternative hemostatic measures. 

## Figures and Tables

**Figure 1 healthcare-10-02523-f001:**
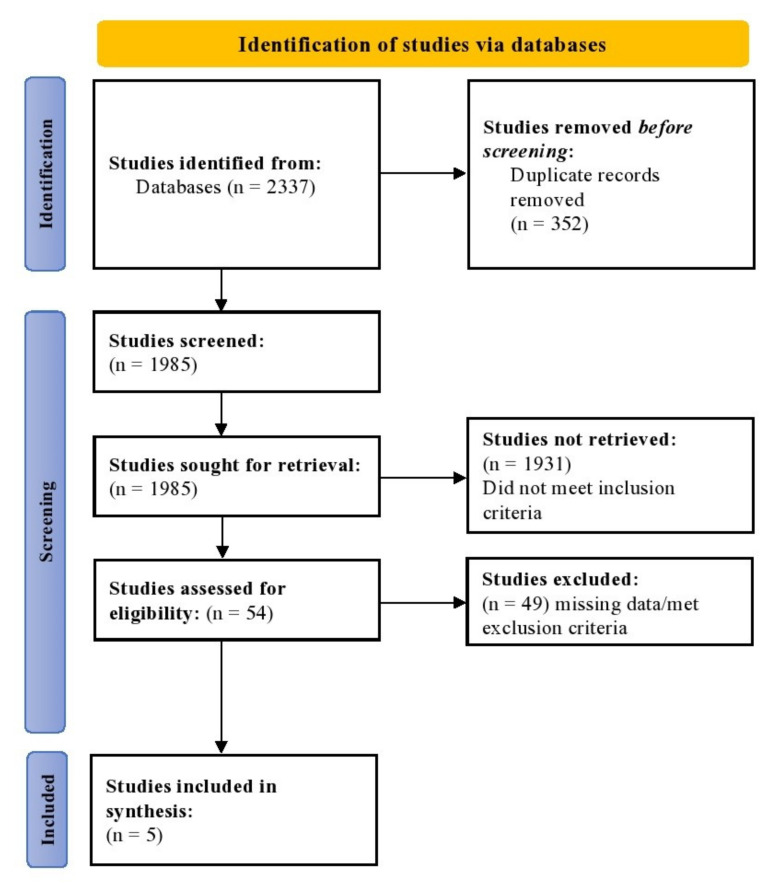
PRISMA flowchart depicting the study selection process. The figure enlists the number of studies identified, the screening process of the studies, and the inclusion process.

**Figure 2 healthcare-10-02523-f002:**
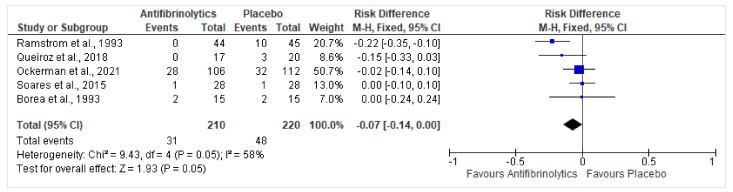
Risk difference (RD) for postoperative bleeding outcomes across the entire cohort of included patients [22,23,24,25,26]. Heterogeneity: Chi^2^ = 9.43, df = 4 (*p* = 0.05); I^2^ = 58%. Test for overall effect: Z = 1.93 (*p* = 0.05).

**Figure 3 healthcare-10-02523-f003:**
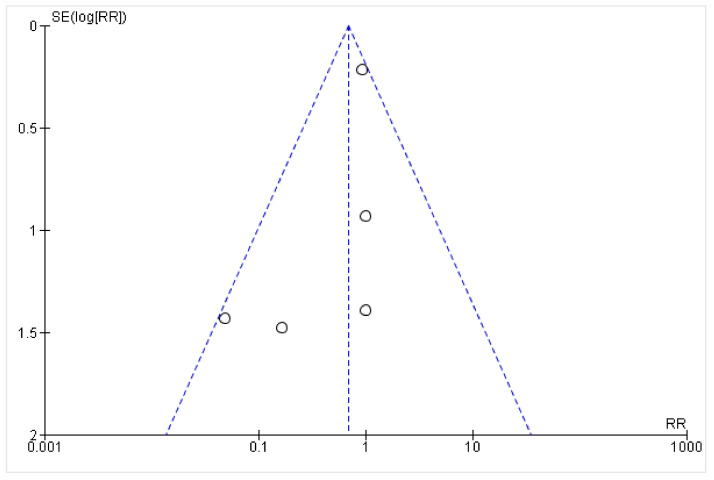
Funnel plot for the visual representation of publication bias.

**Figure 4 healthcare-10-02523-f004:**
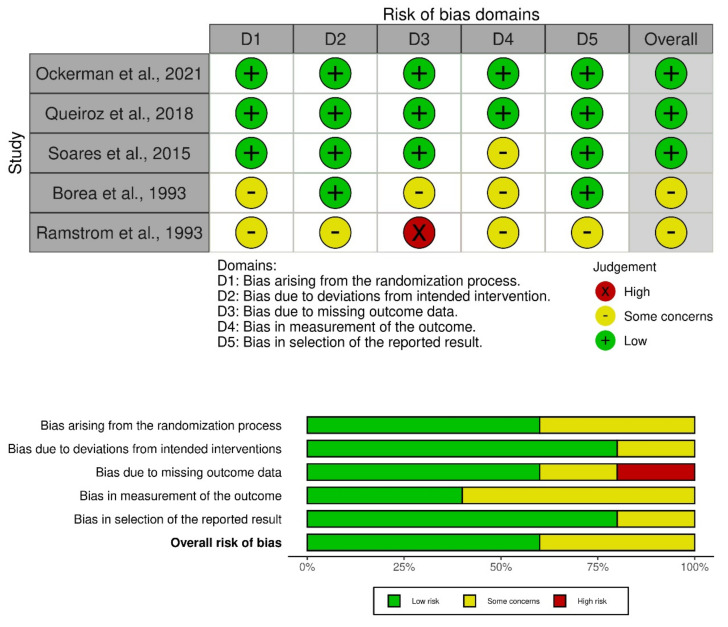
Risk of bias assessment of RCTs using the ROB-2 tool. Traffic light plot of study-by-study bias assessment. Weighted summary plot of the overall type of bias encountered in all studies [9,10,11,12,13].

**Table 1 healthcare-10-02523-t001:** Characteristics of included randomized controlled trials.

Mean Age (Years)		
Author, Year	Intervention Group	Placebo Group
Ockerman et al., 2021 [22]	74.8	72.7
Queiroz et al., 2018 [23]	50.5	41.4
Borea et al., 1993 [24]	62.7	61.1
Ramstrom et al., 1993 [25]	69.8	67.1
Soares et al., 2015 [26]	51	51.1
**Gender (male/female), *n* (%)**		
Ockerman et al., 2021 [22]	Male = 81/106 (76.4%); Female = 25/106 (23.6%)	Male = 64/112 (57.1%); Female = 48/112 (42.9%)
Queiroz et al., 2018 [23]	Male = 8/17 (47.1%); Female = 9/17 (52.9%)	Male = 6/20 (30%); Female = 14/20 (70%)
Borea et al., 1993 [24]	Male = 5/15 (33.3%); Female = 10/15 (66.7%)	Male = 7/15 (46.7%); Female = 8/15 (53.3%)
Ramstrom et al., 1993 [25]	Male = 25/44 (56.8%); Female = 19/44 (43.2%)	Male = 28/45 (62.2%); Female = 17/45 (37.8%)
Soares et al., 2015 [26]	Male = 18/28 (64.3%); Female = 10/28 (35.7%)	Male = 18/28 (64.3%); Female = 10/28 (35.7%)
**Dosage and route of administration of local tranexamic acid**		
Ockerman et al., 2021 [22]	10% tranexamic acid solution mouthwash or placebo solution once before dental extraction and for 1 min 3 times a day for 3 days after dental extraction	
Queiroz et al., 2018 [23]	Two methods of local hemostasis for a minimum of 5 min once and repeated for each group if postoperative bleeding occurs later: (1) IG = irrigation and compression with gauze soaked in 5% tranexamic acid solution and suture, (2) CG = socket irrigation with saline gauze compression only and suture	
Borea et al., 1993 [24]	5% tranexamic acid solution mouthwash (10 mL) or placebo solution after the dental procedure for 2 min 4 times a day for 7 days	
Ramstrom et al., 1993 [25]	Surgical irrigation with 10 mL of 4.8% tranexamic acid solution before suturing followed by 4.8% tranexamic acid solution mouthwash after the dental procedure for 2 min 4 times a day for 7 days, and application of a gauze pad soaked in the solution applied to the bleeding site for 20 min under biting pressure if persistent postoperative bleeding	
Soares et al., 2015 [26]	Three methods of local hemostasis for 8 min once each and repeated for each group if postoperative bleeding occurred later: IG = gauze pad soaked in either 4.8% tranexamic acid solution or fibrin sponge applied to the surgical alveolus for 8 min under biting pressure; CG = dry gauze compression performed under biting pressure on the surgical alveolus for 8 min	
**Indication for anticoagulation in patients**		
Ockerman et al., 2021 [22]	AF = 88 (83%); VTE = 11 (10.4%)	AF = 88 (78.6%); VTE = 14 (12.5%)
Queiroz et al., 2018 [23]	CV = 48.6%; DVT = 40.5%; CVA = 10.8%	
Borea et al., 1993 [24]	All (N = 30; 100%) patients had a CV and were on anticoagulation	
Ramstrom et al., 1993 [25]	Prosthetic CV = 3 (6.5%); VTE = 16 (34.8%); CVD = 6 (13%)	Prosthetic CV = 3 (6.4%); VTE = 13 (27.7%); CVD = 9 (19.6%)
Soares et al., 2015	MV prolapse = 47.4%; Prosthetic CV = 23.7%; VTE = 21.1%; PE = 5.2%	
**Anticoagulation agent used and window before the last dose**		
Ockerman et al., 2021 [22]	Direct oral anticoagulants (rivaroxaban, apixaban, edoxaban, or dabigatran) with an 18–24-h window prior to the dental procedure	
Queiroz et al., 2018 [23]	Warfarin was not suspended prior to the dental procedure	
Borea et al., 1993 [24]	Warfarin was not suspended prior to the dental procedure	Warfarin dose reduced prior to the dental procedure
Ramstrom et al., 1993 [25]	Coumarin drugs (warfarin, dicumarol, or phenprocoumon) not suspended prior to the dental procedure	
Soares et al., 2015 [26]	Warfarin was not suspended prior to the dental procedure	
**Number of patients with postoperative bleeds**		
Ockerman et al., 2021 [22]	28/106 (26.4%)	32/112 (28.6%)
Queiroz et al., 2018 [23]	0/17 (0%)	3/20 (15%)
Borea et al., 1993 [24]	2/15 (13.3%)	2/15 (13.3%)
Ramstrom et al., 1993 [25]	0/44 (0%)	10/45 (22.2%)
Soares et al., 2015 [26]	1/28 (3.6%)	1/28 (3.6%)
**Adverse events due to stopping anticoagulation for dental extraction**		
Ockerman et al., 2021 [22]	0/106 (0%)	1/112 (0.9%)
Queiroz et al., 2018 [23]	3/37 (8.1%) were discontinued in both groups	
Borea et al., 1993 [24]	NR	NR
Ramstrom et al., 1993 [25]	0/44 (0%)	0/45 (0%)
Soares et al., 2015 [26]	0/28 (0%)	0/28 (0%)

AF: atrial fibrillation; CG: control group; CV: cardiac valve; CVA: cerebrovascular accident; CVD: cardiovascular disease; IG: intervention group; MV: mitral Valve; NR: not reported; PE: pulmonary embolism; VTE: venous thromboembolism.

## Data Availability

All data utilized for the purpose of this study are available publicly and online.

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
