# Peer review of "Local Tranexamic Acid for Preventing Hemorrhage in Anticoagulated Patients Undergoing Dental and Minor Oral Procedures: A Systematic Review and Meta-Analysis"

_healthcare, 2022, doi:10.3390/healthcare10122523_

Round 1

Reviewer 1 Report

the present interesting paper evaluate the effect of local tranexanic acid to reduce postoperative bleeding.
The suture or sutureless technique should be cited in the introduction: at line 41 please add the following phrase

"..Further, the use of sutureless technique after tooth extraction influences postoperative bleeding, as well as possibile infection of the wound..."

please cite the following

Chisci G. Sutureless technique in third molar surgery: an overview. J Craniofac Surg. 2013 Nov;24(6):2210-1. doi: 10.1097/SCS.0b013e3182a242ef. PMID: 24220449.

Chisci G, Parrini S, Capuano A. The use of suture-less technique following third molar surgery. Int J Oral Maxillofac Surg. 2013 Jan;42(1):150-1. doi: 10.1016/j.ijom.2012.10.025. Epub 2012 Nov 17. PMID: 23165106.

Chisci G, Capuano A, Parrini S. Alveolar Osteitis and Third Molar Pathologies. J Oral Maxillofac Surg. 2018 Feb;76(2):235-236. doi: 10.1016/j.joms.2017.09.026. Epub 2017 Nov 16. PMID: 29154775.

at line 79, please add the description of the PRISMA flowchart.

In the discussion section the authors should underline possible pharmacological interactions of the tranexanic acid with current drugs in dentistry and report undesired effects.

Author Response

Reviewer 1 Comments and Author Responses:

Comment 1: The present interesting paper evaluates the effect of local tranexamic acid to reduce postoperative bleeding.

The suture or sutureless technique should be cited in the introduction: at line 41 please add the following phrase

"..Further, the use of sutureless technique after tooth extraction influences postoperative bleeding, as well as possible infection of the wound..."

please cite the following

Chisci G. Sutureless technique in third molar surgery: an overview. J Craniofac Surg. 2013 Nov;24(6):2210-1. doi: 10.1097/SCS.0b013e3182a242ef. PMID: 24220449.

Chisci G, Parrini S, Capuano A. The use of suture-less technique following third molar surgery. Int J Oral Maxillofac Surg. 2013 Jan;42(1):150-1. doi: 10.1016/j.ijom.2012.10.025. Epub 2012 Nov 17. PMID: 23165106.

Chisci G, Capuano A, Parrini S. Alveolar Osteitis and Third Molar Pathologies. J Oral Maxillofac Surg. 2018 Feb;76(2):235-236. doi: 10.1016/j.joms.2017.09.026. Epub 2017 Nov 16. PMID: 29154775.

Author Response to Comment 1: Thank you for the much needed input. I have added the sentence under line 41 and have also cited the three studies you listed.

Comment 2: at line 79, please add the description of the PRISMA flowchart.

Author Response to Comment 2: I have added a description of the PRISMA flowchart next to the figure. If you check the paragraph right above the PRISMA figure, a full detailed description of the process is available there.

Comment 3: In the discussion section the authors should underline possible pharmacological interactions of the tranexanic acid with current drugs in dentistry and report undesired effects.

Author Response to Comment 3: To account for your comment, a new subsection has been added in the discussion with the heading: Key Pharmacological Interactions of Tranexamic Acid. Here, we report key interactions and make notes for the field of dentistry. 

Thank you for reviewing the paper and for your helpful insights into improving the paper, exponentially. 

Regards,

Dr. Zouina S.

Reviewer 2 Report

The work is methodologically correct. It is necessary to continue investigating anti-bleeding measures after dental procedures in this type of patient. Only the justification for the female sex being an exclusion criterion is not clear to me.

Author Response

Reviewer 2 Comments and Author Responses:

Comment: The work is methodologically correct. It is necessary to continue investigating anti-bleeding measures after dental procedures in this type of patient. Only the justification for the female sex being an exclusion criterion is not clear to me.

Author Response: Dear reviewer, you may have misread the table where we write “gender/male.” Here we reported the percentage of males, but there are also females in the included studies. Please re-review the table and you will see the female percentage of patients. To clarify, we have included all genders; there were no restrictions. 

Thank you for reviewing the paper and for your helpful insights into improving the paper, exponentially. 

Regards,

Dr. Zouina S.

Reviewer 3 Report

This systematic review with meta-analysis was conducted for an evidence-based decision about the local use of tranexamic acid for preventing hemorrhage in anticoagulated patients undergoing dental and minor Oral Procedures. It is conducted according to the Preferred Reporting Items for Systematic Reviews and Meta-Analyses (PRISMA) statement guidelines.

Minor concerns should be resolved:

1.       Abbreviations under the table should also be included in the text.

2.       The characteristics of included randomized controlled trials should be placed in seven adjacent columns.

3.       Minor English polishing is required.

Author Response

Reviewer 3 Comments and Author Responses:

Comment 1: This systematic review with meta-analysis was conducted for an evidence-based decision about the local use of tranexamic acid for preventing hemorrhage in anticoagulated patients undergoing dental and minor Oral Procedures. It is conducted according to the Preferred Reporting Items for Systematic Reviews and Meta-Analyses (PRISMA) statement guidelines.

Minor concerns should be resolved:

Abbreviations under the table should also be included in the text.

Author Response to Comment 1: Thank you for your comment. The abbreviations have been abbreviated on first-time mentioning and have been abbreviated as follows in the text.

Comment 2: The characteristics of included randomized controlled trials should be placed in seven adjacent columns.

Author Response to Comment 2: On typesetting, the studies when placed in adjacent columns are too long for view hence we placed them vertically. Once processed, the editorial/review team should typeset this best, to my awareness.

Comment 3: Minor English polishing is required.

Author Response to Comment 3: Both myself and my native english speaker/writer colleague have proofread the paper in full. I hope any discrepancies are resolved now.

Thank you for reviewing the paper and for your helpful insights into improving the paper, exponentially. 

Regards,

Dr. Zouina S.

Reviewer 4 Report

- Abstract, line 17: write "and" after the comma

- Introduction and aim: Ok

- Materials and Methods: It is strongly recommended to register the present systematic review in PROSPERO or in another specialized platform, for example, Open Science Framework

- Please, reference the PRISMA 2020 statement

- Line 61: write "and" after the comma

- What did you proceed regarding important missing data from selected articles?

- Data analysis: Did you also use the Q test for assessing homogeneity? Because it is mentioned in the Figure 2

- Risk of Bias assessment: Ok

- Results and Table 1: Ok

- Figure 2: Please, briefly explain the meaning of the heterogeneity results (Chi2 and I2)

- Figure 3: Please, briefly explain how to interpret the funnel plot

- Discussion, Strengths and limitations, and Conclusions: Ok

- Author Contributions: You mention "I.C.O. and Z.S. funding acquisition"; however, below, in Funding, it is stated: "This research received no external funding". Please, clarify 

Author Response

Reviewer 4 Comments and Author Responses: 

Comment 1: Abstract, line 17: write "and" after the comma

Author Response to Comment 1: It has been added.

Comment 2: Introduction and aim: Ok

Author Response to Comment 2: Thank you for your comment.

Comment 3: Materials and Methods: It is strongly recommended to register the present systematic review in PROSPERO or in another specialized platform, for example, Open Science Framework

Author Response to Comment 3: The review’s protocol is both registered and live at Open Science Framework with the following information: osf.io/5rj49 

Comment 4: Please, reference the PRISMA 2020 statement

Author Response to Comment 4: Thank you for your valuable feedback. The PRISMA 2020 statement has been cited.

Comment 5: Line 61: write "and" after the comma

Author Response to Comment 5: It has been added.

Comment 6: What did you proceed regarding important missing data from selected articles?

Author Response to Comment 6: “The biases in missing outcome data pertained to unobtainable information for outcomes of interest in this study.” - This statement has been added to the manuscript as well.

Comment 7: Data analysis: Did you also use the Q test for assessing homogeneity? Because it is mentioned in the Figure 2

Author Response to Comment 7: Yes, it was conducted and it has been updated in the data analysis section.

Comment 8: Risk of Bias assessment: Ok

Author Response to Comment 8: Thank you for your comment.

Comment 9: Results and Table 1: Ok

Author Response to Comment 9: Thank you for your comment.

Comment 10: Figure 2: Please, briefly explain the meaning of the heterogeneity results (Chi2 and I2)

Author Response to Comment 10: A separate brief paragraph has been added to explain the meaning of the findings: “Moderate heterogeneity was present in the included studies with an I2 value of 58%. The Chi2 value was 9.43 (df=4, P=0.05), indicating that there was less variation across the studies as compared to within subjects; the underlying null hypothesis is assumed where the antifibrinolytic treatment effect was the same across studies and variations were simply caused by chance.”

Comment 11: Figure 3: Please, briefly explain how to interpret the funnel plot

Author Response to Comment 11: A separate brief paragraph has been added to explain the meaning of the funnel plot: “On visually inspecting the funnel plot, all five trials were well within the remit of an inverted funnel shape. Each dot represents a single study included in the meta-analysis. The x-axis shows the result of the study whereas the y-axis represents the standard error of the effect estimate. Larger studies are placed towards the top while the lower powered studies are placed at the bottom. With an ideal shape of an inverted funnel or pyramid, the scatter is present due to sampling variation. The shape seen in this funnel plot is expected given the wide range of standard errors. Publication bias may overall be minimal, although our findings must be interpreted with caution.”

Comment 12: Discussion, Strengths and limitations, and Conclusions: Ok

Author Response to Comment 12: Thank you for your comment.

Comment 13: Author Contributions: You mention "I.C.O. and Z.S. funding acquisition"; however, below, in Funding, it is stated: "This research received no external funding". Please, clarify 

Author Response to Comment 13: No funding was acquired for this study. The author contribution statement has been removed for funding acquisition. Thank you for noting the discrepancy.

Thank you for reviewing the paper and for your helpful insights into improving the paper, exponentially. 

Regards,

Dr. Zouina S.

Round 2

Reviewer 1 Report

Manuscript significantly improved; please correct reference 12 citation

Author Response

Reference 12 citation has been fixed.